# Identification of potential biomarkers of vaccine inflammation in mice

Paul F McKay[1†]*, Deniz Cizmeci[1†], Yoann Aldon[1], Jeroen Maertzdorf[2], January Weiner[2], Stefan HE Kaufmann[2], David JM Lewis[3], Robert A van den Berg[4], Giuseppe Del Giudice[5], Robin J Shattock[1]*

[1]Department of Medicine, Division of Infectious Diseases, Section of Virology, Imperial College London, London, United Kingdom; [2]Department of Immunology, Max Planck Institute for Infection Biology, Berlin, Germany; [3]The NIHR Imperial Clinical Research Facility, Imperial Centre for Translational and Experimental Medicine, Imperial College Healthcare NHS Trust, Hammersmith Hospital, London, United Kingdom; [4]GSK, Rockville, United States; [5]GSK, Siena, Italy

*For correspondence:
p.mckay@imperial.ac.uk (PFMK);
r.shattock@imperial.ac.uk (RJS)

†These authors contributed equally to this work

**Abstract** Systems vaccinology approaches have been used successfully to define early signatures of the vaccine-induced immune response. However, the possibility that transcriptomics can also identify a correlate or surrogate for vaccine inflammation has not been fully explored. We have compared four licensed vaccines with known safety profiles, as well as three agonists of Toll-like receptors (TLRs) with known inflammatory potential, to elucidate the transcriptomic profile of an acceptable response to vaccination versus that of an inflammatory reaction. In mice, we looked at the transcriptomic changes in muscle at the injection site, the lymph node that drained the muscle, and the peripheral blood mononuclear cells (PBMCs) isolated from the circulating blood from 4 hr after injection and over the next week. A detailed examination and comparative analysis of these transcriptomes revealed a set of novel biomarkers that are reflective of inflammation after vaccination. These biomarkers are readily measurable in the peripheral blood, providing useful surrogates of inflammation, and provide a way to select candidates with acceptable safety profiles.
DOI: https://doi.org/10.7554/eLife.46149.001

## Introduction

Systems biology approaches are increasingly being used to describe and define signatures of immunity, initially in the setting of infection but more recently in vaccine-induced responses, leading to the development of systems vaccinology (*Querec et al., 2009*; *Olafsdottir et al., 2015*). These transcriptomic analyses have primarily focused on the prediction of vaccine efficacy and immune outcome with the assessment of vaccine safety and potential reactogenicity relying on clinical reactive scores of adverse events (*Bucasas et al., 2011*; *Furman et al., 2013*; *Obermoser et al., 2013*; *Vahey et al., 2010*; *Li et al., 2014*; *Leonardi et al., 2015*; *Vesikari et al., 2009*). However, the transcriptome can also reveal the intricate details of the very early events after vaccination. We have utilized this approach to examine this initial innate response and any subsequent inflammation resulting from a vaccination, to identify biosignatures that have correlative or surrogate potential for vaccine-related inflammation and safety.

It is likely that the activation and maturation of the fundamental responders within the immune system will follow set developmental patterns, which may be revealed by an examination of the transcriptome. Thus any potential differences between each vaccine would probably be due to the degree of response, and to the involvement of different populations of central immune system players and of accessory cells. The dominant determinants of such responses are the nature of the vaccine antigen, its formulation, and the presence or absence of molecularly defined adjuvants

**eLife digest** Measles, whooping cough and other diseases can cause serious illness and death in humans, especially in young children and other vulnerable individuals. Giving people vaccines 'trains' their immune system to recognize and fight the microbes that cause the conditions.

During an infection, the immune system triggers a set of responses that limit the spread of the infectious agent and eliminate it from the body. This can include swelling of tissues (known as inflammation), which in rare cases, can be life threatening.

Inoculations work by sparking a mild immune response in the body. Before a new vaccine is licensed for use, it is thoroughly tested in mice and rodents, and then in human volunteers, to ensure it will cause little or no inflammation. Finding a way to predict early on whether a vaccine candidate will trigger dangerous levels of inflammation would improve this process.

To explore this, McKay, Cizmeci et al. injected the muscle tissue of different groups of mice with one of four licensed vaccines which, by definition, cause little or no inflammation. Other groups of animals were given one of three drugs known to trigger inflammation. Over the following seven days the team repeatedly collected blood as well as cells from the muscle tissue and the lymph nodes. These samples were then analysed to find out which genes were switched on or off at any given time.

The experiments show that the responses of genes in the blood and lymph cells of the mice are connected to those in the muscle cells. Therefore, blood samples may provide a quick and convenient way to assess how an animal is responding to a potential new vaccine. By comparing the genes switched on or off in response to the different vaccines and drugs, McKay, Cizemeci et al. were able to identify a set of genes (known as "biomarkers") that are associated with inflammation in animals. These biomarkers can be used to spot early on whether a new treatment is triggering inflammation.

The next step would then be to identify a similar or identical set of biomarkers in other animals used in vaccine research, and in humans. Ultimately, this approach could make the assessment of the safety of a new vaccine candidate easier.

DOI: https://doi.org/10.7554/eLife.46149.002

(*Li et al., 2017*; *O'Gorman et al., 2015*). The blood and lymphatic systems are the primary routes of transport of cells and chemical signals from one immunological organ to another, with immune cells and soluble mediators trafficking between the injection site and the local draining lymph node (*Girard et al., 2012*; *Jackson, 2014*). Indeed, proteins and molecules that can be measured in the blood after vaccination are likely to be useful biomarkers of such responses as blood draws can be readily taken from most individuals. Although a particular cell or inflammatory molecule may be present in relatively high numbers and concentrations in both the lymph node and the injection site, it is apparent that the large volume of blood in the circulatory system may dilute this molecule and prevent detection.

In this study, we examined the molecular signatures generated in the blood, the draining lymph node and the injection site in the muscle of mice after the administration of a number of licensed vaccines, which set the benchmark for what is safe, and of a set of known agonists of Toll-like receptors (TLRs) or adjuvants, to set an upper boundary of inflammatory response (*Haziot et al., 1998*; *Qureshi et al., 1999*; *Akira and Takeda, 2004*; *Fitzgerald et al., 2004*; *Matsumoto and Seya, 2008*). The four licensed vaccines used in this study have a substantial history of safe use in humans, with large datasets supporting their safety and tolerability. They ranged from protein alone to vaccines that included adjuvants or a known component that have an acceptable degree of reactogenic potential (whole cell pertussis) (*Frey et al., 2003*; *Tregnaghi et al., 2012*; *Mohanty et al., 2018*; *Van Den Ende et al., 2017*). We also used the water-in-oil emulsion incomplete Freunds adjuvant (IFA) (Montanide ISA 51) and two TLR agonists (lipopolysaccharide (LPS) and Poly I:C) (*Table 1*). A set of indicators of vaccine-elicited inflammation in a small animal model will provide an early warning system that can be used to inform decisions on the potential tolerability and ultimately the utility of vaccines undergoing selection and development, and will provide additional tools to de-risk late-stage failure.

**Table 1.** Vaccines and inflammatory agents.

The vaccines, TLR agonists, adjuvant and saline used in the present study are detailed. Vaccines were administered by injection of 1/10th the human dose in a 50 μL volume. TLR agonists, IFA and saline were also given in a single 50 μL volume. All injections were into the mouse hind leg quadriceps muscle.

| Vaccine/ TLR | Components | Abbreviation | Manufacturer |
|---|---|---|---|
| Pentavac SD | Diphtheria, tetanus, pertussis (whole cell), hepatitis B (rDNA) and haemophilus type b conjugate vaccine | Pentavac | Serum Institute India |
| Agrippal | Trivalent flu subunits – H3N2, H1N1 and influenza B | Tri-Flu | Seqirus |
| Fluad | Trivalent flu subunits – H3N2, H1N1 and influenza B + MF59 | Tri-Flu + MF59 | Seqirus |
| Engerix B | Recombinant hepatitis B sAg absorbed on alum | | GlaxoSmithKline |
| IFA | Montanide ISA 51 VG | | Seppic |
| LPS | LPS-EB Ultrapure | | Invivogen |
| Poly I:C | Polyinosinic:polycytidylic acid | | Sigma |

DOI: https://doi.org/10.7554/eLife.46149.003

# Results

## Transcriptomic profiles in muscle, draining lymph nodes and PBMCs of mice treated with different licensed vaccines and TLR agonists

High-quality RNA samples, isolated from muscle tissue excised around the injection site, the draining medial iliac lymph nodes (MLN) and total peripheral blood mononuclear cells (PBMCs), were taken at 4, 8, 24, 48, 72, 168 hr after the vaccination or treatment with LPS, polyI:C or IFA (*Table 1*) and subjected to genome-wide transcriptome analysis. The unvaccinated mice, and mice receiving saline alone, were used as control groups.

Differentially expressed genes were defined as those with a significant (Benjamini-Hochberg (BH) adjusted p-value<0.01) change in expression in a vaccinated group when compared to the unvaccinated group and the saline group at a given time point. Violin plots that chart the numbers of upregulated and downregulated differentially expressed genes revealed that the immunisations differed in terms of the magnitude and the kinetics of the transcriptomic responses induced in muscle, MLN, and blood (*Figure 1*). Strong responses were observed in the injected muscle tissue for all of the immunisations, with the highest levels of fold change and of absolute numbers of differentially expressed genes being detected at 72 and 168 hr after treatment. Pentavac SD, Poly I:C and particularly LPS also induced changes in differential gene expression in the muscle, MLN and blood at 4, 8, 24 and 48 hr post-injection. Engerix B and IFA elicited modest changes in the differential gene expression in the muscle at 48, 72 and 168 hr and at the later time points of 72 and 168 hr in the MLN, but very few differentially expressed genes were observed in the blood. Agrippal and Fluad exhibited similar patterns of differential gene expression at the later time points of 72 and 168 hr, but Fluad had an additional earlier signal at 8, 24 and 48 hr in the muscle (*Figure 1*).

## Distinctive transcriptional programs elicited after immunisation: gene-set enrichment and co-expression analysis

Gene-set enrichment analysis was performed to identify the functional patterns of the biological processes that are induced by the different immunisations. Gene modules that are specific to the context of immune responses in blood tissue were previously defined by *Li et al. (2014)*. These blood transcriptional modules were used to assess the signatures induced by the immunisations at the different time points. The blood transcriptional module enrichment profiles for every tissue are compared and presented in *Figure 2* and *Figure 2—figure supplements 1–3*, where the gene sets with significant enrichment (lower than $p<10^{-6}$) in each tissue are shown. In the injected muscle tissue, each vaccine or TLR agonist (with the exception of Agrippal) primarily elicited an upregulation of expression, as compared with the expresson levels seen after injection of the saline control, of genes related to inflammation, growth factors, innate immunity and cell damage (*Figure 2—figure*

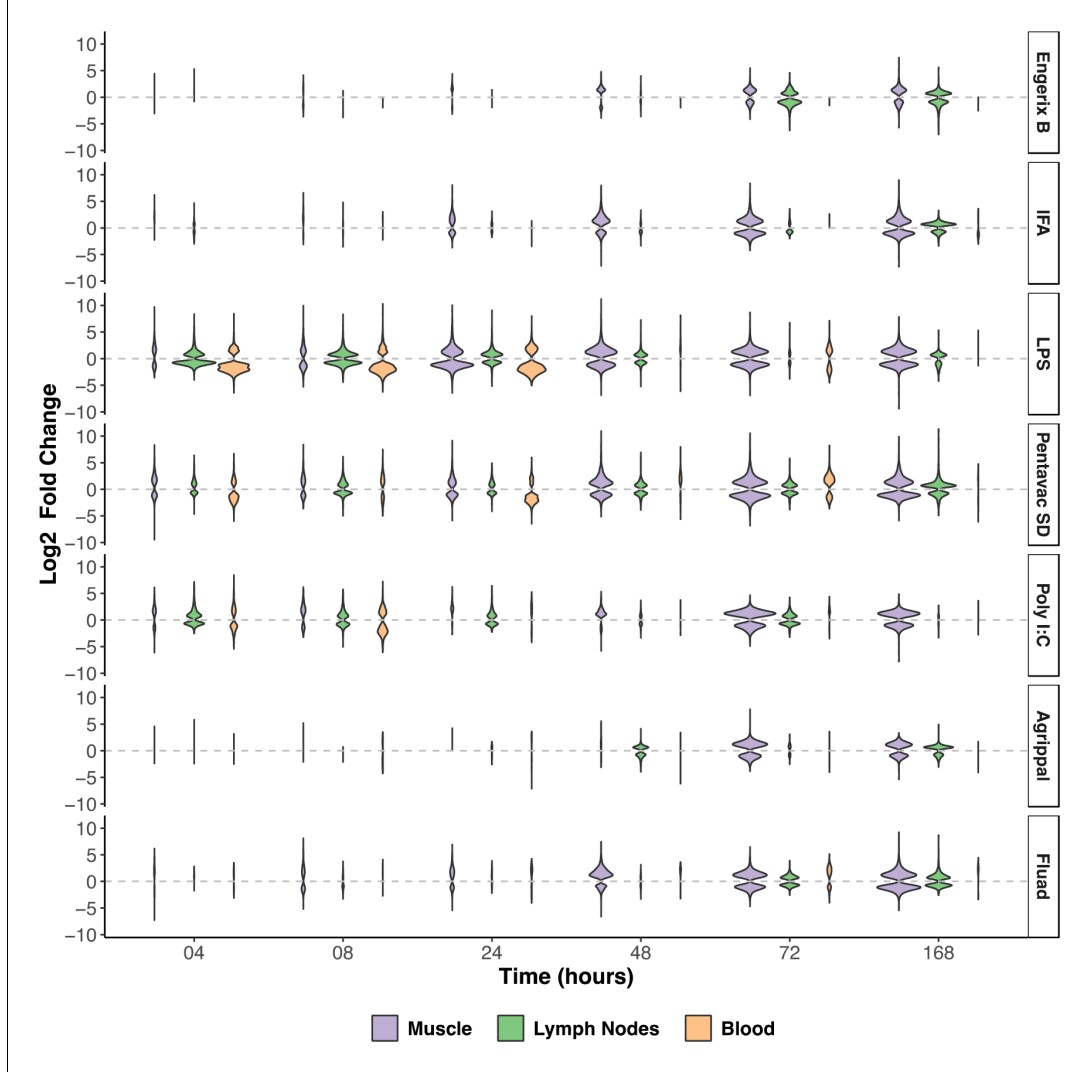

**Figure 1.** Overview of the transcriptomic responses to the injected vaccines and TLR agonists in the injected muscle, the draining lymph nodes (MLN), and the blood. The width of each violin is scaled to the number of genes that are differentially expressed at each given time point (BH-adjusted p-value <0.01). Differentially expressed genes were defined as those with a significant (BH-adjusted p-value <0.01) change in expression in a vaccinated group, when compared to the unvaccinated group and the saline group at a given time point. The upregulated or downregulated differentially expressed genes (Log2 fold-change) are presented for each tissue sampled and at each timepoint after injection of the vaccine or TLR agonist (n = 5 mice per timepoint). The vaccines or TLR agonists that were used are indicated on the right of the figure. IFA, incomplete Freunds adjuvant; LPS, lipopolysaccharide.

DOI: https://doi.org/10.7554/eLife.46149.004

supplement 1). The induced transcriptomic response in the MLN was more refined, probably reflecting the effect of the migration of activated immune cells and local generalised stimulation by cyto/chemokine gradients. In these draining iliac lymph nodes, the injection of LPS, Pentavac SD and Poly I:C promoted an elevation of the interferon response genes as well as of cyto/chemokines (*Figure 2—figure supplement 2*). Finally, in the PBMCs, the transcriptomic analysis revealed that LPS, Pentavac SD and Poly I:C elicited an increase in gene transcripts that was associated with the interferon response and with activated monocytes and neutrophils (*Figure 2—figure supplement 3*).

To identify transcriptional patterns in an unbiased way, gene modules were constructed using Weighted Gene Co-expression Network Analysis (WGCNA) (*Langfelder and Horvath, 2008*). Briefly, WGCNA discovers interacting genes by computing and matching the correlation patterns in their expression profiles over the entire time period, extracting clusters (modules) of highly correlated genes, and thereby categorizing the association between the modules and the traits (different

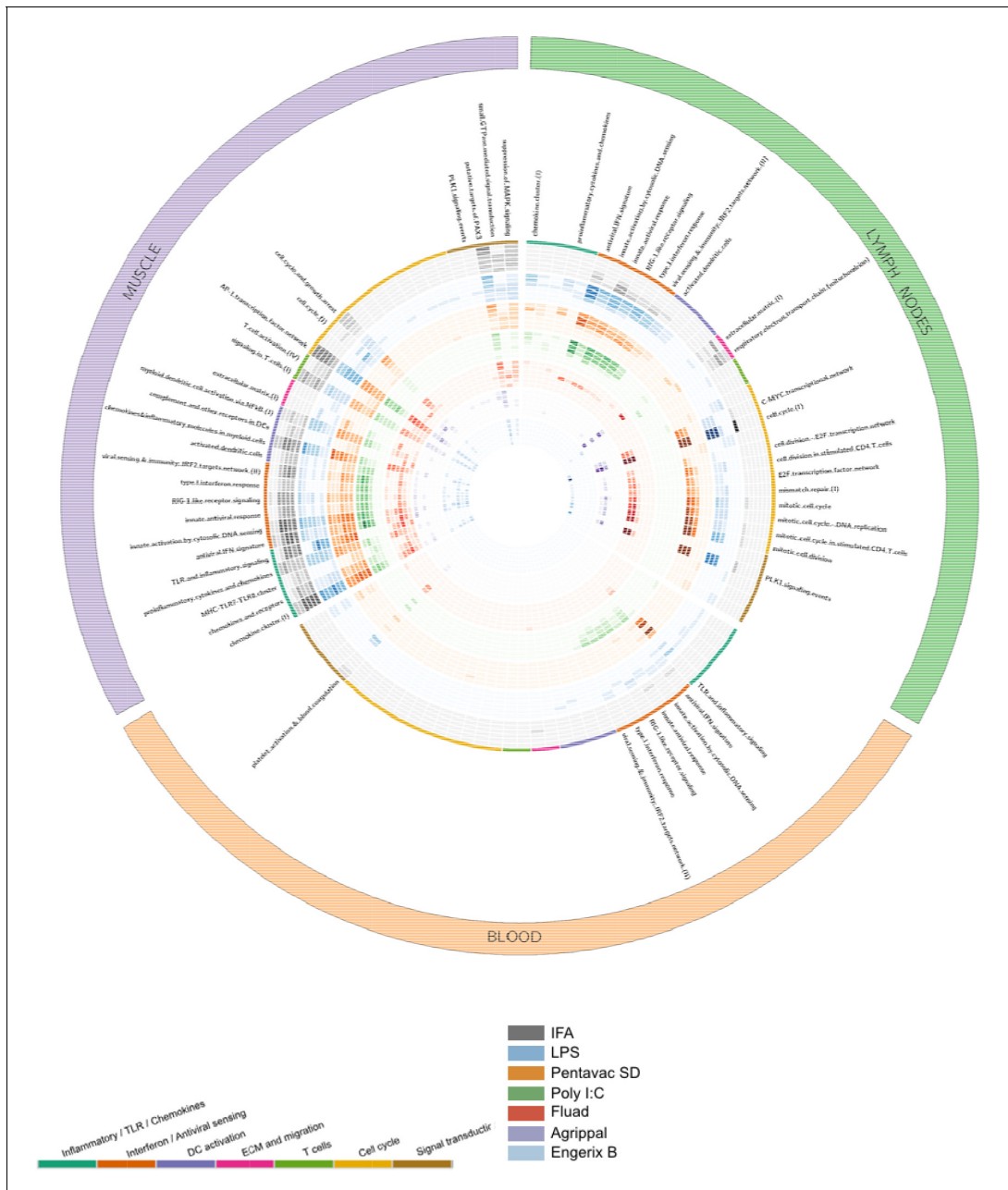

**Figure 2.** Gene-set enrichment analysis reveals distinct mechanisms of transcriptional response to the injected vaccines and TLR agonists. Tissue data sets are presented as three coloured segments along the perimeter for the injected muscle (purple), the draining lymph nodes (green), and the blood compartment (red). An ordered list of blood transcriptional modules is labelled inside the outermost perimeter. Coloured bands adjacent to the module names correspond to the vaccines and Toll-like receptor (TLR) agonists (outside to inside bands): incomplete Freunds adjuvant (IFA) (grey), lipopolysaccharide (LPS) (blue), Pentavac SD (orange), Poly I:C (green), Fluad (red), Agrippal (purple), and Engerix (blue). For each band, there are six rings corresponding to time points 4, 8, 24, 48, 72, 168 hr (from outer to inner ring). The intensity of colours indicates the significance of the enrichment. Only modules with p-value for enrichment of $<10^{-6}$ and an effect size (AUC) >0.8 are shown. DC, dendritic cell; ECM, extracellular matrix.

DOI: https://doi.org/10.7554/eLife.46149.005

The following figure supplements are available for figure 2:

**Figure supplement 1.** Gene-set enrichment analysis in the muscle data set.
DOI: https://doi.org/10.7554/eLife.46149.006

**Figure supplement 2.** Gene-set enrichment analysis in the draining lymph nodes data set.
DOI: https://doi.org/10.7554/eLife.46149.007

**Figure supplement 3.** Gene-set enrichment analysis in blood data set.

*Figure 2 continued on next page*

*Figure 2 continued*

DOI: https://doi.org/10.7554/eLife.46149.008

vaccines). WGCNA was applied to the gene expression datasets for muscle, lymph nodes, and blood separately. The co-expression analyses identified 40 modules in muscle with numbers of genes ranging from 27 to 4979 (*Figure 3—figure supplement 1A*), 47 modules in lymph nodes with numbers of genes ranging from 26 to 3456 (*Figure 3—figure supplement 1B*), and 30 modules in blood with numbers of genes ranging from 24 to 9789 (*Figure 3—figure supplement 1C*). WGCNA assigns colours to name each module. We also added 'mu' for muscle, 'ln' for lymph nodes, or 'bl' for blood to each module name to avoid modules having the same name across different tissue datasets (such as muscle module (muM1_turquoise), lymph node module (lnM1_turquoise) and blood module (blM1_turquoise)). *Figure 3—figure supplement 1* shows the correlations (r >0.3 positive correlations are in blue, r <−0.3 negative correlations are in red) between the modules and the immunisations. In muscle, co-expression analysis was able to identify modules that correlated with each immunisation, strongly (both positive and negative) for Pentavac SD and to a lesser degree for LPS, Fluad and Agrippal. These modules were different to those that correlated with a saline-only injection. In lymph nodes and blood, the method again revealed that Pentavac SD, LPS, Fluad and Agrippal treatments were associated with a number of positively and negatively significantly correlated modules. The connections between these highly correlated genes modules, which reveal whether the same sets of genes are co-regulated in the different tissues, are shown in *Figure 3—figure supplement 1D*, which identifies consensus modules associated with Agrippal, LPS and Pentavac SD.

To interpret the extracted modules functionally, we compared the WGCNA modules with the reference blood transcriptional modules. A hypergeometric test was performed to quantify the overlap between the two sets of modules and the significant enrichments are reported in a heatmap (*Figure 3*). In muscle, the muM1-turquoise module contains genes that overlap with most of the reference modules and is associated positively with LPS and Pentavac SD; the muM2-blue module overlaps with reference modules of extracellular matrix (ECM), migration and mitochondria and is negatively associated with LPS and Pentavac SD (*Figure 3* and *Figure 3—figure supplement 1A*). In lymph nodes, the lnM6-red module exhibits significant overlap with the reference modules associated with the cell cycle, whereas the lnM8-pink module is associated with interferon or antiviral sensing and this module contains genes that are co-regulated in LPS and Poly I:C immunisation (*Figure 3* and *Figure 3—figure supplement 1B*). In blood, the blM1-turquoise module overlaps with reference modules of T cell, whereas the M4-yellow module overlaps with reference modules of monocytes, neutrophils and inflammatory/TLR/chemokines. The M1-turquoise and M4-yellow modules contain genes of similar transcriptional patterns that are significantly associated with LPS and Pentavac SD immunisation (*Figure 3* and *Figure 3—figure supplement 1C*).

We developed an interactive web interface (available at https://vaccinebiomarkers.com) to facilitate data access and further discovery. This website allows users to (1) query genes and visualise their transcriptional profiles for each condition, (2) filter the differentially expressed genes by their functional groups and visualise the fold changes, and (3) analyse WGCNA modules by visualising the functional enrichments and listing the genes of each module.

## Identification of biomarkers that reflect potential vaccine inflammation

In order to reveal potential connections between the different tissues, the differentially expressed genes and each sampling time point, we created a circular heatmap diagram showing genes that were common between the three tissues (*Figure 4*). We first selected the top 100 genes that were differentially regulated in both the injected muscle tissue and the MLN, then we assessed whether any of these genes were also differentially regulated in the circulating PBMC's RNA expression profiles. This analysis identified a set of genes that coded for soluble or cell-associated proteins present in each of the compartments that were differentially regulated by immunisation. We focused on soluble markers to simplify the sampling and quantification of potential blood biomarkers. The outside rim of the circular heatmap indicates the common individual genes, and the strength of the correlation for each tissue and each immunisation treatment is colour- and sized-coded. (Black indicates positive correlation, red indicates negative correlation; the thickness of the lines

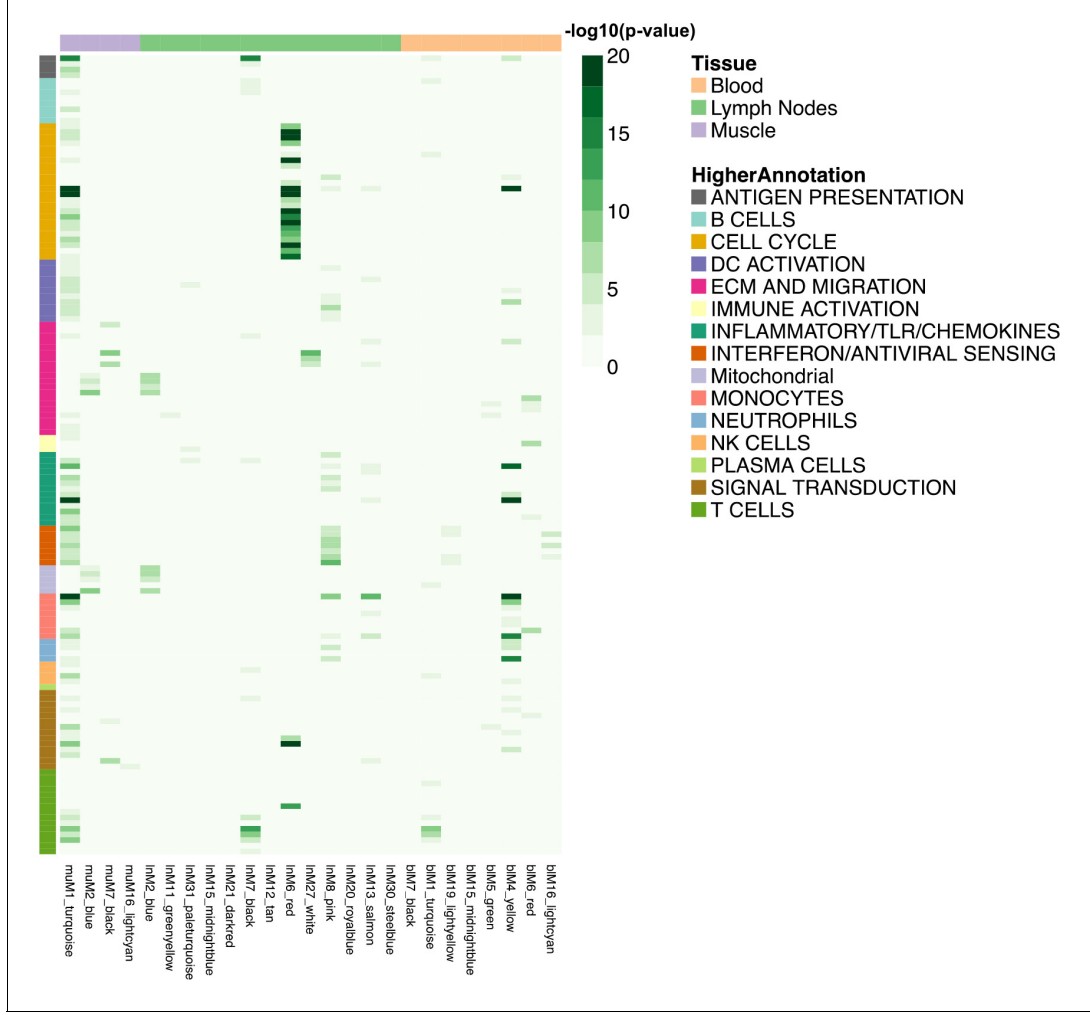

**Figure 3.** Heatmap representing the transcript overlap between WGCNA modules and the reference modules. Reference modules are the blood transcriptional modules defined by *Li et al. (2014)*. These modules are shown in rows and are annotated within a higher functional group. Only WGCNA modules that showed a significant enrichment (hypergeometric test adjusted p-value<0.05) in any of the reference modules were included and shown in columns (hypergeometric test adjusted p-value<0.05). The legends on the figure report the strength of the p-value as a gradient of green. The colours at the top of the columnsindicate the tissue being analysed, whereas rows are colour-coded to indicate their higher annotation, as indicated in the key.

DOI: https://doi.org/10.7554/eLife.46149.009

The following figure supplement is available for figure 3:

**Figure supplement 1.** Weighted gene co-expression network analysis of module–trait relationships.

DOI: https://doi.org/10.7554/eLife.46149.010

corresponds to the magnitude of the correlation coefficient.) Many of these genes encoded chemokines and/or cytokines but the analysis also identified proteins that collectively have been termed acute-phase proteins. These proteins are typically present at high levels during inflammatory events, and they include serum amyloid A-3 (SAA3) and pentraxin 3 (PTX3). Murine SAA3 is an ortholog of the human SAA3 pseudogene and is involved in the murine response to bacterial endotoxins, often acting in combination with TLR2 (*Ather and Poynter, 2018*; *He et al., 2009*), whereas the long pentraxin PTX3 facilitates pathogen recognition by macrophages and dendritic cells (*Diniz et al., 2004*). SAA3 was strongly induced in muscle tissue after most of the immunisations (the exception being Agrippal, whereas in the draining lymph node and the PBMC transcriptome, only LPS, Pentavac SD and poly I:C enhanced RNA expression levels. LPS caused the greatest alteration in gene expression profiles of cyto/chemokine genes in the blood, enhancing CCL2, CCL3, CCL4, CXCL1, CXCL2,

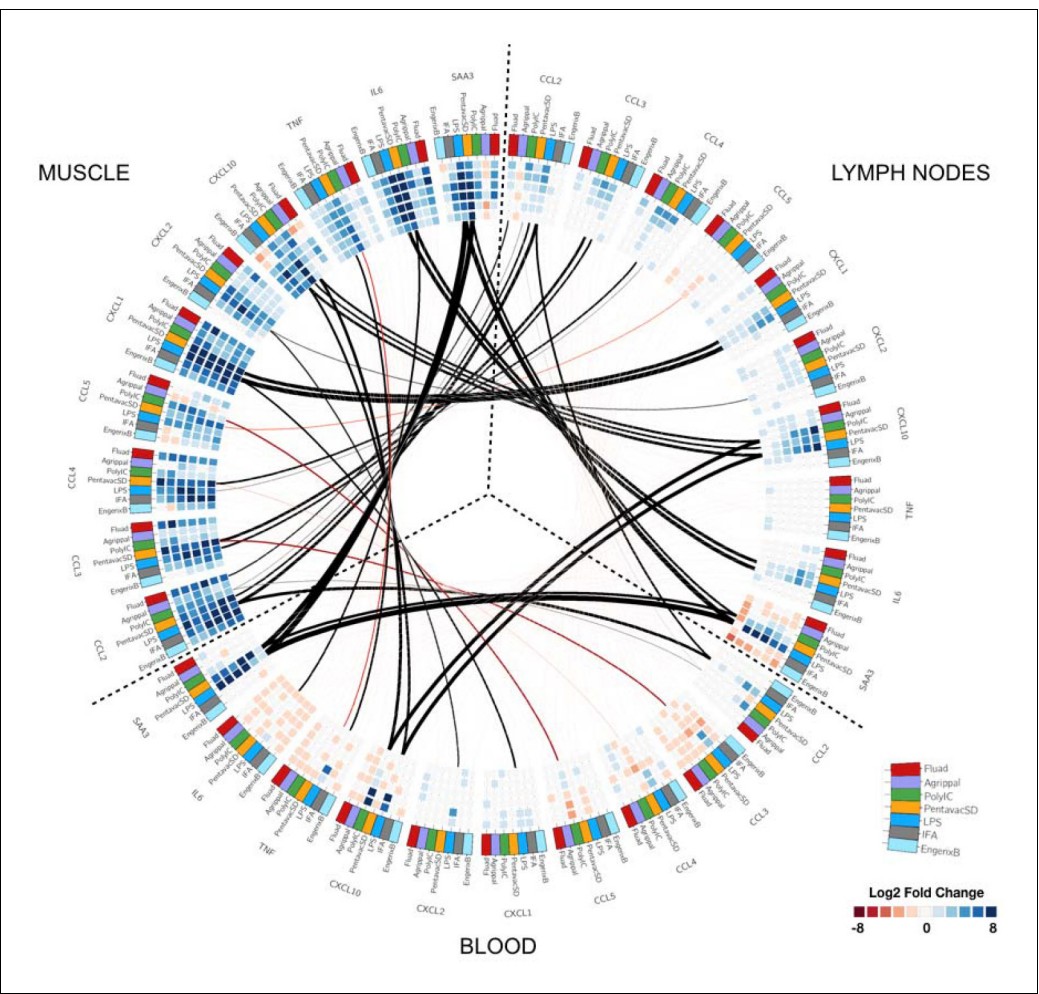

**Figure 4.** Circular map representing the differentially expressed genes and reporting the observed correlations between each tissue. The three tissues analysed are organised as in *Figure 2* and are indicated on the outer part of the circular map. Each outer tile has a different colour to denote which injection they received: Fluad (red), Agrippal (purple), Poly I:C (green), Pentavac SD (yellow), LPS (blue), IFA (grey) and Engerix B (light blue). Each cell in the inner circles shows the fold change for each given time point, treatment, and tissue (dark blue/blue for upregulation and dark red/red for downregulation, with the density of the colour corresponding to the magnitude of fold change). The lines connecting the different subsets of tiles represent the correlation between tissues for each gene-treatment combination with black lines showing a positive correlation and red lines a negative correlation. The thickness of the lines indicates the magnitude of the correlation.

DOI: https://doi.org/10.7554/eLife.46149.011

CXCL3, CXCL9, CXCL10, CXCL13 and TNF-alpha expression but not affecting CXCL5. These differences were seen mainly at the very early time points of 4 and 8 hr and were not maintained at 24 hr post intramuscular immunisation with the LPS. Notably, in the draining lymph node, the TNF-alpha levels are not elevated. In the muscle, there is strong general upregulation of IL-6 and elevated expression of IL-6 also occurs in the lymph node after LPS and Pentavac SD immunisation, whereas there is a generalised downregulation of IL-6 in the PBMCs. The licensed vaccines, surprisingly including the Pentavac SD which contained the whole cell pertussis, induced minor perturbations of the RNA expression levels in the PBMC. Poly I:C strongly upregulated CCL2 and CXCL10 and mildly enhanced CXCL1, CXCL9 and CXCL13 in the blood, and interestingly this pattern was slightly different from the expression in the draining lymph node where CXCL9, CXCL10 and CCL2 were enhanced while CXCL1 and CXCL13 were unaffected (*Figure 4*).

## Identification of biomarkers in the serum

We next examined whether a marker identified from the transcriptomics could be measured in the blood. We measured a panel of cyto/chemokines by Luminex and SAA3 by ELISA in sera harvested from mice vaccinated with saline control, the two licensed vaccines (Pentavac SD and Fluad) and the two potent TLR agonists (LPS and Poly I:C). We selected these treatments on the basis of the range of responses observed from the transcriptomic analysis. Pentavac SD, LPS and Poly I:C had signatures of cytokine responses in the MLN, Fluad much less so, but such responses were still detectable, suggesting that these were good candidates for enabling cyto/chemokine detection in the peripheral blood sera (*Figures 2* and *4*). Serum was collected from five mice per itreatment and per analysis time point. LPS was clearly inflammatory at early time points after injection, eliciting strong expression of CCL2, CCL3, CCL4, CCL5, CXCL1, CXCL2 and CXCL10.Following treatment with LPS, TNF-α and IL-6 proteins were significantly above baseline and saline control levels at 4 and 8 hr, with a rapid return to much lower expression levels by 24 hr and to basal levels by 48–72 hr (*Figure 5A*). Levels of CXCL10 were still significantly above those in controls at 24 hr after LPS injection, having reached a peak at 4 hr after treatment which then declined to almost half by 8 hr but was still significantly ($p<0.05$) above controls and baseline at 503 pg/mL by 24 hr post immunisation. In addition, polyI:C elicited measurably elevated levels of a number of these cyto/chemokines, specifically CXCL10 and CCL5 at 4 hr (CXCL10 – 6,263 pg/mL; CCL5 – 8,727 pg/mL, $p<0.0001$) and 8 hr (CXCL10 – 1,462 pg/mL; CCL5 – 3,642 pg/mL, $p<0.0001$) and CCL2 and CCL4 at 4 hr (CCL2 – 3,944 pg/mL; CCL4 – 1,249 pg/mL, $p<0.0001$). Strikingly, all treatments apart from the saline control elicited very high levels of expression of the SAA3 protein, which were at least 1000-fold greater than those of any other measured analyte, and moreover the kinetics of SAA3 expression were also of a longer duration than those for expression of the cyto/chemokines (*Figure 5A*). The TLR4 agonist LPS elicited the highest peak SAA3 response of 492.8 µg/mL at 24 hr ($p<0.0001$), which reduced considerably by 48 hr (99.74 µg/mL, $p<0.0001$), reaching a level of 5.96 µg/mL by 72 hr and baseline levels after 168 hr. Although the Pentavac SD vaccination did not achieve the same peak level of SAA3 expression as the LPS injection, the levels of SAA3 continued to increase until 48 hr post-immunisation with Pentavac SD (332.6 µg/mL, $p<0.0001$) and were maintained until 72 hr (245.6 µg/mL, $p<0.0001$), remaining significantly above baseline levels at the final analysis time point of 168 hr (104.8 µg/mL, $p<0.0001$). A comparison of the total accumulation of SAA3 after the LPS or Pentavac SD vaccinations revealed that the AUC for Pentavac SD was more than twice that of LPS, being 31,455 µg.hr/mL and 14,572 µg.hr/mL, respectively. The TLR3 agonist Poly I:C invoked an SAA3 expression profile in which the molecule reached 33.11 µg/mL at 4 hr before falling back to 15.04 µg/mL at 8 hr and rising again to 33.54 µg/mL at 24 hr, although these differences did not reach statistical significance when compared to the saline control. Interestingly, treatment with Fluad, which contains the oil-in-water emulsion adjuvant MF59, did not generate a peak in the expression level of SAA3 in the sera until 24 hr post-injection (81.82 µg/mL, $p=0.0014$), suggesting that a delayed mechanism of action is induced by this emulsion. *Figure 5B* shows the fold changes over saline alone of measured proteins that are induced by different vaccines at the different time points, giving an indication of the degree and duration of expression over the background levels. In the case of the LPS immunisation, this analysis showed that an 'expression set' of cyto/chemokines and SAA3 can be defined to include CCL2, CCL3, CCL4, CCL5, CXCL1, CXCL2, CXCL10, IL-6 and TNF-α but not SAA3 at 4 hr, then the same set of cyto/chemokines but including SAA3 at 8 hr. These comparisons of the differential transcriptomic expression in the blood with the actual levels of expressed proteins measurable in the animal sera revealed that many of the proteins closely matched. The strong expression of CCL2, CCL3, CCL4, CXCL2, CXCL10, and TNF-α proteins that was elicited by LPS immunisation was in line with the measured transcriptomic changes at early time points (4, 8 and 24 hr). By contrast, there is a downregulation in transcript levels for CCL5, but CCL5 protein levels are elevated at 4, 8, and 24 hr. The sustained upregulation of SAA3 following the Pentavac SD vaccination was reflected in both transcript and protein levels.

## Correlation between transcripts and circulating cytokines

We next quantified the correlations between blood transcript and protein fold changes across all time points for all measured cyto/chemokines in the LPS immunisation group (*Figure 5—figure supplement 1*). CXCL1, CXCL10, and SAA3 showed strong correlations that were statistically significant

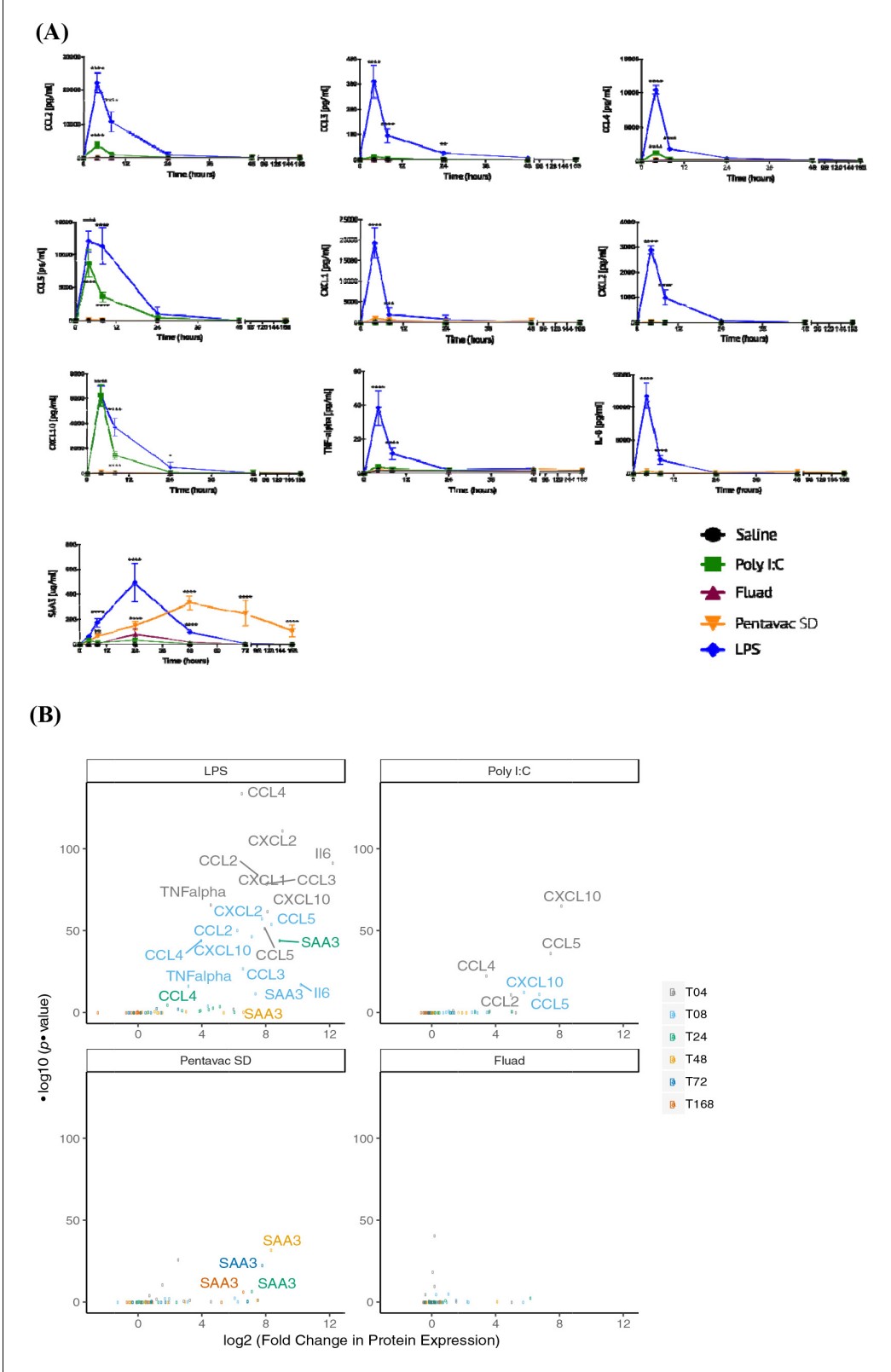

**Figure 5.** Serum chemokine and cytokine expression. Serum chemokines and cytokines were measured using a Luminex bead 9-plex assay, and serum amyloid A3 was measured by quantitative ELISA. (**A**) Each panel shows the longitudinal expression of an individual chemokine/cytokine in murine peripheral blood sera at times 0, 4, 8, 24, 48, 72 and 168 hr post-injection with the indicated vaccine (n = 5 mice per time point). Chemokine/cytokine levels

*Figure 5 continued on next page*

*Figure 5 continued*

induced by different vaccines that were in excess of those induced by saline alone were analysed using two-way ANOVA followed by Dunnett's multiple comparisons test, **p<0.01, ***p<0.001, ****p<0.0001. (**B**) Volcano plot representing the differential expression of all measured proteins in the sera induced by different vaccines shown as fold-change in the levels induced by saline alone at different time points. Data points corresponding to responses with adjusted p-value <0.01 are labelled with the protein name, and colour coded according to the time point.

DOI: https://doi.org/10.7554/eLife.46149.012

The following figure supplement is available for figure 5:

**Figure supplement 1.** Comparison of gene expression levels measured by microarray and protein levels of chemokines and cytokines measured by Luminex bead 9-plex assay and quantitative ELISA.

DOI: https://doi.org/10.7554/eLife.46149.013

(r >0.6, p-value <0.05). Although LPS generated the highest levels of cytokine expression, the transcriptomic data also revealed different patterns of differential gene expression for the other treatments. To reveal any potential relationship between the differential gene expression in the different tissues and the protein signal that is detected in the serum, we compared the differential gene expression for each vaccine, from each tissue, over the whole period of study and correlated these gene expression levels to the serum protein expression levels at the same time points (*Figure 6*). Each vaccine or TLR agonist had different correlation profiles. The correlation between gene expression in the lymph node and blood and the blood serum proteins was most pronounced after LPS injection, with only the differential expression of CCL5 in the blood and the level of CCL5 serum protein being significantly negatively correlated. The muscle tissue had only one significant correlation with blood SAA3 levels in the LPS condition. By contrast, after Poly I:C injection, we identified a significant correlation between IL-6 transcript levels in the muscle tissue and IL-6 protein in the blood, although a number of other blood proteins correlated well but not significantly with the differential expression of their respective genes. Interestingly, the Fluad vaccine produced the highest number of significant correlations between gene expression in the muscle and the proteins present in the blood, indicating that most of the biomarker proteins present in the sera after Fluad vaccination may be derived from the injected muscle tissue. The Pentavac SD vaccine induced no significant correlations between the differentially expressed genes in the blood and the proteins in the blood, correlations were primarily noted between changes in gene expression levels in the MLN, and partially from the injected muscle, and serum protein levels.

## Discussion

Despite the widespread use of vaccines and an enormous quantity of data from both animals and humans, the development, degree and mechanism of vaccine sensitivity or potential reactogenicity are still unclear. This is perhaps not surprising because the immune system is a complex network in which robust regulation to prevent run-away activation or self-attack is essential. We used a systems vaccinology approach combining multi-tissue transcriptomics with a simple serum analysis to define a set of biomarkers for potential vaccine-related inflammation in mice. We examined the transcriptomic profile of the injected muscle tissue, the draining lymph nodes and the peripheral blood after injection of four well-tolerated and safe licensed vaccines. These included a safe vaccine that is known to elicit moderate reactogenicity in many people because it contains whole cell *Bordetella pertussis*, as well as a number of molecularly defined TLR agonists that should generate significant inflammatory reactions. The added value of examining the transcriptomic response in the injected muscle tissues and then in the draining lymph node, and then finally determining whether there is a cellular signature in the blood, was that we could potentially follow the development of the response through these compartments. Of course, there will be cross-talk between these compartments because cells and inflammatory mediators travel from the injection site into the blood and to the lymph nodes and visa-versa, either early on as an indicator of an innate response to a danger signal (e.g. vaccines plus any adjuvant) or later as an indicator of cells arriving at the site of injection in response to innate signals generated in the muscle tissue and adaptive responses in the draining lymph node.

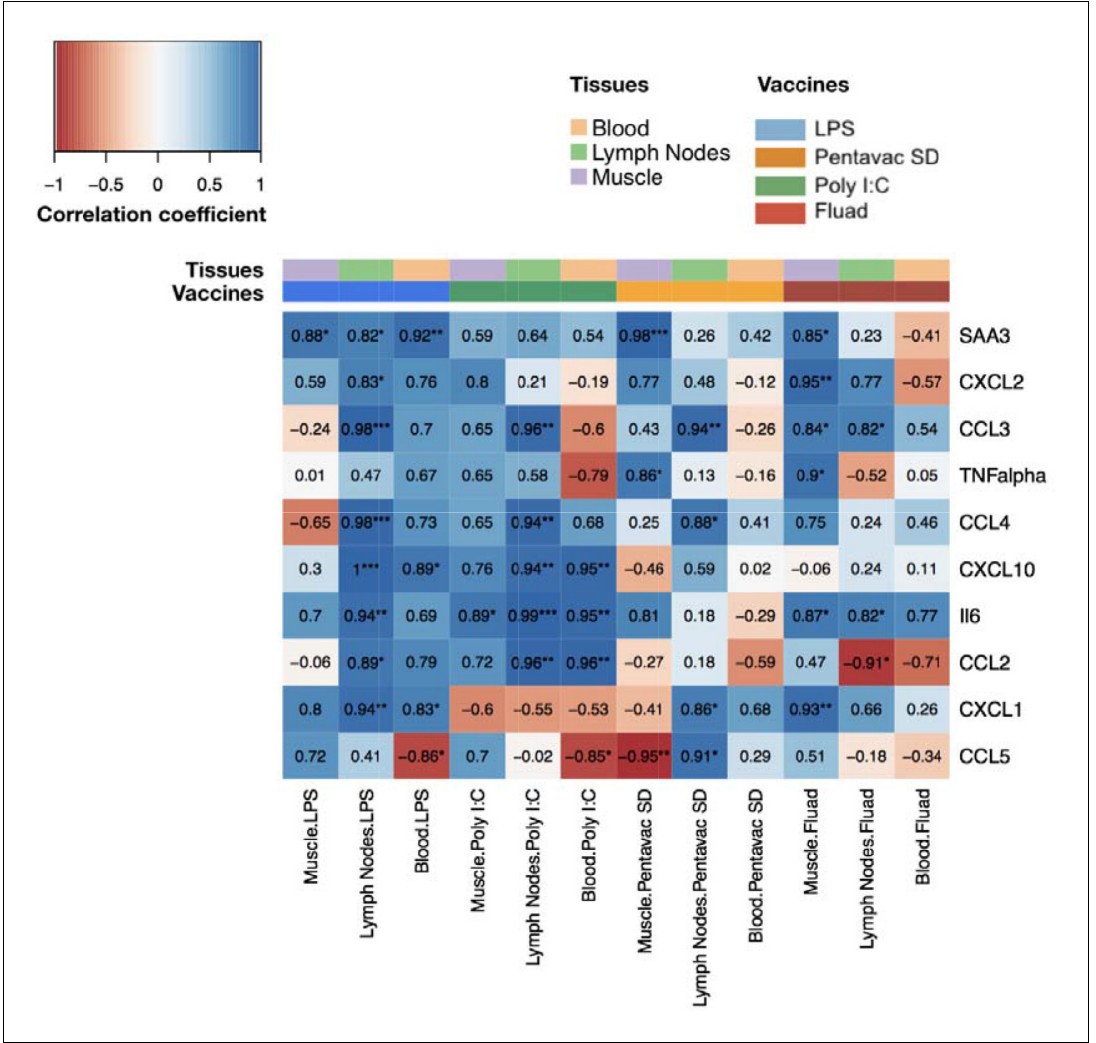

**Figure 6.** Comparison of gene expression levels measured by microarray with the protein levels of chemokines and cytokines measured by Luminex bead 9-plex assay and quantitative ELISA. The correlations between fold changes in transcript and protein levels are presented in this heatmap. All measured chemokine and cytokines are in rows (left axis legend). Columns represent different tissues and immunisations ,as indicated by the colour-coded legend above the heatmap. Each cell shows the coefficient of correlation between protein levels in murine peripheral blood sera and the transcript changes measured at the same time points(4, 8, 24, 48, 72 and 168 hr after vaccination) in the muscle, lymph nodes, and blood. Fold changes of protein levels were calculated at each time point by dividing the mean values of proteins in each vaccine group by the mean values of those of the saline-alone group. Correlations between the fold changes in transcript and protein levels were calculated using as Pearson correlation coefficients. Each cell shows the correlation coefficient followed by a sign for the significance with *p<0.05, **p<0.01 and ***p<0.001 (see legend on the top left corner).
DOI: https://doi.org/10.7554/eLife.46149.014

From the violin plots that visualize the absolute numbers of genes that have significant differential expression when compared to the saline control or unvaccinated animals, it is clear that the Engerix B, Agrippal or Fluad vaccines produce a very low signal in any of the three compartments, the peripheral blood, the draining lymph nodes or the injected muscle. This is reassuring given that these are licensed vaccines with substantial history of safe use. However, for three investigational conditions — injection with the licensed vaccine Pentavac SD (a safe but more reactogenic vaccine), with the TLR4 agonist LPS, or with the TLR3 agonist Poly I:C — there were a large number of differentially expressed genes that were measurable in the muscle, lymph nodes and the peripheral blood. Interestingly, these differences were particularly evident at the very earliest time points

of 4 and 8 hr after immunisation, with many differentially expressed genes returning to baseline levels by 24 hr post immunisation. These extremely rapid changes are difficult to capture in studies investigating human responses to vaccination, with much published work only looking at transcriptomic changes at days 1, 3 or 7 or after a much longer time period because the focus of their studies was to investigate and correlate the transcriptomic responses with the development and maturation of an adaptive immune response (*Querec et al., 2009*; *Li et al., 2014*; *Kazmin et al., 2017*; *Haks et al., 2017*; *Davis et al., 2017*). In order to rationalize the large datasets generated by transcriptomic analysis, we used a gene set enrichment technique to identify groups of genes that were assigned to specific groups on the basis of the blood transcriptional modules defined by *Li et al. (2014)* and that were significantly enriched over the saline-alone injection. Our analyses revealed that the injected muscle tissue exhibited a large number of transcriptionally enriched modules, interestingly focused mainly on the innate and adaptive immune response, dendritic cells and T cells. As expected, the innate-sensing modules were elevated at earlier time points whereas the cell cycle module enrichments came later, mainly after 48 hours post injection. The response to Poly I:C was notable because this injection caused a significant enrichment of innate modules, but this did not follow through to generate any downstream cell-cycle response.

Lower signals in the blood could be due to a number of potential reasons. Genes or sets of genes may be expressed only in the muscle, only in the draining lymph node or indeed only in the blood. The blood signature could also be a downstream result of the increased expression of a gene in the muscle or lymph node tissues, or alternatively might be caused directly by the systemic dissemination of the injected material. Alternatively, there could be an element that is expressed strongly in both the muscle and draining lymph node that we don't see in the blood due to dilution of the responding cells in the large number of cells in the circulation, compared to, for example, a concentrated accumulation of responsive cells or expanding clones in the lymph node. A primary aim of this work was to identify a signal in the peripheral blood that reflected events occurring in the injected muscle site and/or in the draining lymph nodes; therefore, we next looked at potential connections between the different tissues. An analysis that created a circular heatmap diagram showing the interconnections between the three compartments revealed a strong positive correlation between the injected muscle and the lymph nodes for CCL3, CCL4, CXCL1 and IL-6, with a much weaker connection for these molecules with the peripheral blood compartment. However, there was a positive correlation between the upregulation of CCL2, CXCL10 and SAA3 gene expression in the tissues and the blood compartment, with the latter being particularly high in the first 48–72 hr.

We then examined the serum protein levels of specific chemokines, cytokines and SAA3 to determine the relationship between the level of the transcript and the protein product. The expression of transcripts can be downregulated while the proteins are still being made, a level of control that isn't unexpected, particularly for highly pleiotrophic proteins such as cyto/chemokines that need to be carefully regulated to prevent dysregulation of the immune and other systems (*Kovarik et al., 2017*). It is also possible that high levels of transcripts may not result in high levels of protein because of the modulation of translation in highly active or stressed cells (*Grootjans et al., 2016*). Luminex and ELISA analysis of the serum samples surprisingly showed that for many of the cyto/chemokines, the levels of proteins peaked within 4 hr of injection, returning to baseline levels by 24–48 hr post injection, with clear differences in the patterns of expression between the different treatments. LPS was by far the most potent inducer of all of the cyto/chemokines measured, with only Poly I:C eliciting similar levels of CCL5 and CXCL10. In the case of the reactive protein SAA3, both LPS and Pentavac SD caused a marked increase in the protein levels in the sera, with expression kinetics that were very different from those of cyto/chemokine expression and in quantities that were 100–1000-fold higher than those of the cyto/chemokines. When graphed in a volcano plot, showing the highly significant fold-change differential expression of the proteins, it is clear that groups of cyto/chemokines and SAA3 allow us to differentiate specific signatures for each individual vaccine condition.

A comparison of blood transcript and serum protein data revealed significant and informative correlations. For LPS vaccination, there was a strong relationship between the differential gene expression of CXCL1, CXCL10 and SAA3 and the corresponding proteins in the blood. However, a number of other cyto/chemokine proteins that were significantly elevated in the serum did not correlate with the expression of differentially regulated genes in the blood, posing the question of the source of the protein products of these differentially expressed genes. A further comparison of potential

correlations revealed that there were marked relationships between differentially expressed genes in the muscle and draining lymph nodes and the serum proteins, indicating that chemokines produced in the muscle and lymph nodes were released into the circulating blood.

In the current study, we have identified a set of biomarkers, or biological indicators, that are measurable in the peripheral blood and that reflect an inflammatory response resulting from an injection into a distal muscle. The biomarker identity was nuanced, with different immunisations generating identifiably distinct levels of serum biomarker proteins. These serum biomarkers were reflective of events in the injected muscle tissue, the draining lymph nodes or the peripheral blood. Our analysis demonstrates that the serum biomarker samples should be taken soon after immunisation, certainly within the first 12 hr, because as the signature is much reduced by 24 hr and an early sampling will be both practically convenient and will provide an early signal of adverse reactions. By looking only at those serum proteins that exhibited clear differentiation between the licensed and safe vaccines and those that were correlated with inflammation caused by the tested TLR agonists, we can create a panel of recommended biomarkers of potential inflammation. This panel would include SAA3, CCL2 and CXCL10, with the presence and expression level of all three of these proteins being indicative of adverse or potentially harmful inflammation. The presence of an acute-phase reactive protein such as SAA3 is perhaps not surprising, although these proteins are primarily produced by hepatocytes (*Tannock et al., 2018*). Interestingly, this specific SAA-family member was identified through our analyses as being differentially regulated in the transcriptomes of each of the three tissues, whereas the more highly expressed SAA1 and SAA2 were not. Assessment of these biomarkers of potential inflammatory signals in future pre-clinical studies may give an early indication of unwanted or excessive inflammation, and could potentially identify vaccine candidates that have harmful or highly inflammatory profiles. With the aim of facilitating future discovery, we have created an interactive web interface as a tool for further interrogation and multivariate analysis of these data, which is accessible at https://vaccinebio-markers.com.

The mouse is typically the initial animal model used in early vaccine development and, rightly or wrongly, has traditionally functioned as an immunogenicity gatekeeper for the progression of vaccine candidates into higher animal models and ultimately to human clinical trials. Studies examining comparative transcriptomics in mice and men have variously described mouse models as having either a very poor or very high and significant correlation with human responses. Indeed a pair of articles famously utilized the same data sets to demonstrate both sides of this debate (*Seok et al., 2013*; *Takao and Miyakawa, 2015*). The published datasets from these studies, though quite different from those in our current study in terms of the time points analysed, tissues examined and interventions used, did include an injection of LPS via the intraperitoneal route. An examination of the differential gene expression datasets after the LPS treatment in the *Takao and Miyakawa (2015)* work revealed that CXCL10, CCL3 and CCL4 were among the top 50 positively regulated genes. SAA3 was not part of this dataset (*Seok et al., 2013*; *Takao and Miyakawa, 2015*). Several studies have examined the transcriptomic signature of MF59 in blood, draining lymph nodes and injected muscle tissue (*Mosca et al., 2008*; *Liang et al., 2017*; *Francica et al., 2017*). Transcriptomic analysis of mouse muscles injected with MF59 revealed a picture that is strikingly similar to that produced by our current study, in which significant fold-change elevations were noted for SAA3, CCL2, CCL4, CCL5 and CXCL10 (*Mosca et al., 2008*). A similar study that was also performed in mice, although using a different oil-in-water squalene emulsion (AS03), demonstrated changes in the gene expression levels that were similar to those observed in our study for the Fluad vaccination (*Morel et al., 2011*). The CCL2, CCL3, CCL4, CXCL1, CXCL10 and IL-6 cyto/chemokines were significantly elevated at 4 and 24 hr post treatment in the injected muscle and also in the draining lymph nodes.

Our identification of CCL2 and CXCL10 as potential markers of vaccine-elicited inflammation is therefore consistent with previously published studies. These chemokines are likely to play a significant role in the context of generating effective vaccine-elicited immunity as both are critical modulators of leukocyte trafficking and homing to the site of vaccine injection. CCL2 is a key chemokine regulating the movement of monocytes and macrophages, memory T cells and NK cells through vascular endothelium and the infiltration of the tissue into which the vaccine or inflammatory mediator has been applied (*Sozzani et al., 1993*). CCL2 is readily produced by muscle cells in response to damage, inflammation or infection and potently recruits monocytes and macrophages upon

engagement of the highly expressed CCR2 receptor, concomitantly setting up an autocrine amplification of the CCL2 chemokine in the monocytes and macrophages (*Cushing et al., 1990*; *Yoshimura et al., 1989*; *Chiu et al., 2012*). This CCL2 amplification may further enhance inflammation at the site of injection and potentially the perceived immunological danger that a vaccine will lead to a subsequently higher level of immune responses. CCL2 has also been reported to influence the polarisation of developing antigen-specific T cells towards a Th2 phenotype, indicating that the presence of CCL2 in the cyto/chemokine milieu may influence the resulting vaccine-elicited T cell immunity (*Karpus et al., 1997*). CXCL10 was first described as an interferon-induced chemokine that is produced by a wide range of cell types, including monocytes and dendritic cells (*Ciesielski et al., 2002*; *Ge et al., 2012*; *Ohmori and Hamilton, 1995*; *Holm et al., 2012*). CXCL10 binds to CXCR3 and promotes immune cell trafficking and homing to inflamed tissues as well as the perpetuation of inflammation in a similar fashion to CCL2. Likewise, CXCL10 has reported effects on T cell immunity and is critical for the generation of the protective CD8 T cell responses induced by activated dendritic cells. Both CCL2 and CXCL10 are potent recruitment signals for several of the key players in the generation of an immune response, monocytes or macrophages, neutrophils and dendritic cells. The presence of these cells at the site of injection has been shown to augment antigen-specific B and T cell immune responses significantly (*McKay et al., 2004*; *Sumida et al., 2004*; *Calabro et al., 2011*).

Our current study builds upon previous publications by comparing responses to four licensed vaccines with extensive safety and tolerability profiles to those produced by highly immunostimulatory TLR agonists and the IFA adjuvant. Furthermore, we performed a longitudinal transcriptomic analysis of the injected muscle site, the draining lymph nodes and the peripheral blood, as well as an analysis of proteins in the sera in order to begin to define the optimum time and the biomarkers that can be used to measure inflammatory responses after immunisation. While we have identified a set of inflammatory biomarkers that are applicable to mice, more work is needed to determine whether this set or a set of human biosignatures will act as potential signifiers of vaccine inflammation in humans.

# Materials and methods

## Key resources table

| Reagent type (species) or resource | Designation | Source or reference | Identifiers | Additional information |
|---|---|---|---|---|
| Strain, strain background) *M. musculus*, Female) | CB6F1/Crl; Strain Code 176 | Charles River | | |
| Commercial assay or kit | RNAprotect animal blood tubes | Qiagen | Cat number: 76544 | |
| Commercial assay or kit | miRNeasy mini kit | Qiagen | Cat number: 217004 | |
| Commercial assay or kit | QIAzol lysis reagent | Qiagen | Cat number: 79306 | |
| Commercial assay or kit | 9-plex Luminex | Bio-Techne | Cat number: LXSAMS-09 | |
| Chemical compound, drug | Pentavac SD | Serum Institute India | | |
| Chemical compound, drug | Agrippal | Seqirus | | |
| Chemical compound, drug | Fluad | Seqirus | | |
| Chemical compound, drug | Engerix B | GlaxoSmithKline | | |
| Chemical compound, drug | IFA - Montanide ISA 51 VG | Seppic | NSC#737063 | |

*Continued on next page*

*Continued*

| Reagent type (species) or resource | Designation | Source or reference | Identifiers | Additional information |
|---|---|---|---|---|
| Chemical compound, drug | LPS-EB Ultrapure | Invivogen | Cat number: tlrl-3pelps | |
| Chemical compound, drug | Poly I:C | Sigma | Cat number: P1530 | |

## Ethics statement

The animal studies were approved by the Ethical Review Board of Imperial College London, where the experiments were carried out and work was performed in strict compliance with project and personal animal experimentation licences granted by the UK government in accordance with the Animals in Scientific Procedures Act (1986) – PPL 70–7457 Protocol #1. Animals received minimal handling and their physical condition was monitored at least twice daily. All procedures were performed under isoflurane anaesthesia when appropriate, and all efforts were made to minimize suffering. There was a detailed protocol in place, as required by the humane endpoints described in the animal licence, for early euthanasia in the event of onset of illness or significant deterioration in condition. At the end of the experiment, all animals were culled using a schedule one method and death confirmed before necropsy. Food and water were supplied ad libitum.

## Animals, immunisation and sampling

Female CB6F1 mice of 6–8 weeks of age were purchased from Charles River. Animals received a single injection in their right hind leg quadricep muscle and were then culled at specific intervals after the immunisation, either 4, 8, 24, 48, 72 or 168 hr. A control group that did not receive an immunisation was culled and tissues harvested. A further control group received a single saline injection in the right hind leg quadriceps muscle and was then culled at the same intervals as the animals that received an active formulation. When each animal was culled, the injected muscle site and the iliac lymph nodes that drain the hind leg quadriceps muscles were harvested and flash frozen in liquid nitrogen. Peripheral blood, sampled from the mouse tail vein immediately before humane euthanasia, was collected (100 µL) in RNAprotect animal blood tubes (Qiagen, UK). Groups of mice (n = 5 per group per time point) received 1/10th of the human dose in 50 µL of one of the following licensed vaccines: Pentavac SD (diphtheria, tetanus, pertussis (whole cell), hepatitis B (rDNA) and *Haemophilus influenzae* type b conjugate vaccine) (Serum Institute India, Pune, India); Agrippal (trivalent flu subunits – H3N2, H1N1 and influenza B) (Novartis Vaccines, now Seqirus, UK); Fluad (trivalent flu subunits – H3N2, H1N1 and influenza B + MF59 (oil-in-water emulsion)) (Novartis Vaccines, now Sequirus, UK); Engerix B (recombinant hepatitis B surface antigen absorbed on aluminium) (GSK, Rixensart, Belgium), or either Poly I:C (Sigma, UK – P0913: 50 µL of a 1 mg/mL solution), LPS (Invivogen, UK – LPS-EB Ultrapure: 50 µL of a 0.5 mg/mL solution), IFA (Seppic, France – Montanide ISA 51 VG: 50 µL of a 1:1 mixture of IFA and Saline), or saline alone (Sigma, UK – 50 µL). The 1/10th of a human dose received by the mice was based on the 'mouse equivalent dose'. This estimation takes into account various measures and differences between animal species including the body surface area and metabolic rate and is an FDA accepted method for dose conversion (*Sharma and McNeill, 2009*).

## Total RNA preparation

### Tissue samples

Small pieces of tissue (3 mm x 3 mm x 3 mm) were harvested and flash frozen in liquid nitrogen. Total RNA isolation (including microRNA (miRNA) species) was performed using the miRNeasy mini kit (Qiagen, UK), as described in the standard protocol for purification of miRNA and total RNA from tissues and cells. Briefly, 700 µL QIAzol lysis reagent was added to the tissue sample which was then disrupted and homogenized using a tissue homogenizer (tissueruptor) probe. Homogenate was incubated at room temperature (RT) (15–25°C) for 5 min before addition of 140 µL chloroform and vigorous shaking for 15 s. After incubation at RT for 2–3 min, the homogenate was centrifuged for 15 min at 12,000 x *g* at 4°C. The upper aqueous phase was then transferred to a new collection tube (350 µL). 1.5 volumes (525 µL) of 100% ethanol were added and mixed thoroughly by pipetting then

transferred into a RNeasy Mini column, the liquid being pulled through the column by vacuum manifold. The RNA on the column was washed using 700 µL Buffer RWT, followed by 2 washes using 500 µL Buffer RPE. The RNeasy Mini column was then placed into a new 2 mL collection tube and centrifuged at full speed for 1 min to dry the membrane completely. The RNeasy Mini column was subsequently transferred to a new 1.5 mL collection tube and 30 µL RNase-free water was directly pipetted onto the column membrane, and centrifuged for 1 min at $\geq$8000 x $g$ to elute the purified RNA, which was stored at $-80°$C until required for microarray hybridisation.

## Blood samples

100 µL of peripheral blood harvested from the mouse tail was collected directly into an RNAprotect Animal Blood Tube and the tubes incubated at RT (15–25°C) for 2 hr. The blood tubes were then flash frozen in liquid nitrogen for storage. RNA isolation was performed as described in the kit protocol for purification of Total RNA, including miRNA, from RNAprotect stabilized animal blood tubes (100 µL). Briefly, completely thawed tubes were first centrifuged for 3 min at 5000 x $g$, then the supernatant removed by careful pipetting. The pellet was resuspended in 1 mL of RNAase-free water and transferred to a clean RNAse-free 2 mL tube. The pellet was completely resuspended in the water, centrifuged for 3 min at 5000 x $g$ and the supernatant carefully removed by pipetting. The pellet was then dissolved in 240 µL Buffer RSB before addition of 200 µL Buffer RBT and 20 µL proteinase K enzyme solution. The tube was vortexed for 5 s then incubated at 55°C for 10 min in a shaking incubator at 1,200 rpm. After incubation, the sample was added into a Qiashredder spin column and centrifuged for 3 min at 10,000 x $g$. The supernatant was then transferred to a RNase-free tube and 690 µL 100% ethanol added, before being mixed thoroughly by pipetting. Then, the sample was transferred into a RNeasy Mini column, and pulled through the column by vacuum manifold. The sample on the column was washed using 350 µL Buffer RWT, followed by the addition of 80 µL DNase I solution (in Buffer RDD) and incubation at RT for 15 min. After DNase I treatment, the column was washed with a further 350 µL of RWT followed by two washes using 500 µL Buffer RPE and a final wash with 500 µL 80% RNase-free ethanol. The RNeasy Mini column was then placed into a new 2 mL collection tube and centrifuged at full speed for 1 min to dry the membrane completely. The column was then transferred into a new 1.5 mL collection tube and 30 µL RNase-free REB buffer was directly pipetted onto the column membrane, then centrifuged for 1 min at $\geq$8000 x $g$ to elute the purified RNA, which was stored at $-80°$C until required for microarray hybridisation.

## Whole-genome microarray analysis

Gene expression data were generated from high-quality RNA samples on an Agilent microarray platform (Agilent Technologies). RNA was labelled with a Low Input Quick Amp Labeling Kit (Agilent Technologies) according to the manufacturer's instructions. Quantity and labelling efficiency were verified before hybridization to whole-genome 8 $\times$ 60 k mouse expression arrays (Agilent design ID 028005), and scanned at 5 µm using an Agilent scanner. Image analysis and data extraction were performed with Agilent's Feature Extraction software (version 11.5) to generate the raw expression data. The complete set of microarray data was deposited in the NCBI's Gene Expression Omnibus and is accessible through GEO accession number GSE120661.

## Transcriptomic analysis

Data analysis was performed in R version 3.3.2 (2016-10-31). Microarray data were pre-processed, normalised and analysed for differential expression using R package limma v3.28.14 (*Ritchie et al., 2015*). The raw data were first background corrected using the *normexp* method. Background corrected signals were quantile normalised between arrays. Linear models were fitted using the limma *lmFit* function. All treatments and time points were part of a single model and separate models were fitted for each tissue. There was an unvaccinated group for each treatment. Contrasts were designed to compare each of the different stimulus groups to unvaccinated animals and also to the saline control at each time point, using the interaction term (treatment.time point – treatment.unvaccinated) – (saline.time point – saline.unvaccinated). Differential expression was evaluated using the moderated *t*-statistics and the *p*-values were adjusted using Benjamini and Hochberg's (BH) method (*Benjamini and Hochberg, 1995*). The violin plots visualizing the strength of the transcriptomic

responses were created using ggplot2 geom_violin, in which areas are scaled proportionally to the number of differentially expressed genes (adj. p-value<0.01).

Genes that are orthologs in mice and humans were assigned using NCBI HomoloGene (*US National Library of Medicine, 2004*). Gene set enrichment analysis was performed with R package tmod (version 0.34) using CERNO statistical test (*Weiner and Domaszewska, 2016*; *Yamaguchi et al., 2008*). We calculated p-values corrected for multiple testing using the BH procedure and the effect size area under curve (AUC) of the gene set enrichment for blood transcriptional modules (BTMs) defined by *Li et al. (2014)*. BTMs were assigned to high-level annotation groups using the annotations defined by *Kazmin et al. (2017)*. Circular visualizations of the functional modules were performed using Circos version 0.69.4 (*Krzywinski et al., 2009*). Highly concordantly as well as highly discordantly regulated genes between tissues were identified using the method described by *Domaszewska et al. (2017)*. Magnitude of gene expression change (effect size), significance (adj. *p*-value) and direction of gene expression change are used to determine the discordance/concordance score (disco.score).

Weighted gene correlation network analysis (R package WGCNA version 1.51) was used to find clusters of highly correlated genes among the stimulus groups (*Langfelder and Horvath, 2008*). Signed co-expression networks were constructed using the Pearson correlation as the similarity measure and the minimum module size was set to 20. The first principal component of the expression matrix (module eigengene) of each constructed module is calculated using the *moduleEigengenes* function of the WGCNA package. Sample trait is provided as a binary indicator variable of the immunisation status. The Pearson correlations of the module eigengenes with traits were calculated to determine the association between several modules of co-expressed genes and the administered vaccines. We used BTMs to reveal the functional roles of the constructed WGCNA modules. A hypergeometric test was used to test the enrichment of co-expressed genes with genes specific to the BTMs defined by *Li et al. (2014)*. To find modules that are shared between muscle, lymph nodes, and blood networks, a consensus module analysis was carried out. Modules are connected on the basis of the number of genes they have in common and are visualized in a network diagram using the force-directed algorithm in R package *igraph* v1.0.1.

## Luminex and ELISA analysis of mouse sera

Groups of mice (n = 5 per group per time point) received 1/10th of the human dose in 50 µL of one of the licensed vaccines (*Table 1*). We analysed the sera of a subset of the conditions that were used for transcriptomics. These were selected on the basis of the differential gene expression analyses that had identified significant differences in the profiles and that also had the potential for these differences to be observed in the peripheral blood. Sera from immunised mice (time points: 4, 8, 24, 48, 72 and 168 hr) and baseline sera obtained from naïve mice (n = 5) at time 0 hr were analysed using either a 9-plex Luminex or a Mouse Serum Amyloid A ELISA (Bio-Techne, Abingdon, UK), according to the manufacturer's protocols. Briefly, the Luminex filter-bottomed microplate was first pre-wet with 100 µL wash buffer then placed on the vacuum manifold to remove the buffer through the filter. 50 µL of the microparticle bead cocktail was added to each well followed by 50 µL of the diluted standard or sample and the mixture was incubated for 2 hr at RT on a horizontal orbital shaker set at 500 rpm. Wells were washed 3x by the addition of 100 µL wash buffer, with each wash being drawn through the filter membrane using the vacuum manifold. 50 µL of diluted biotin antibody cocktail (specific for each analyte) was added to each well and the microparticle beads were incubated with the biotinylated antibody mixture for 1 hr at RT on the orbital shaker at 500 rpm. The plates were washed again using 3 × 100 µL wash buffer per well, and then 50 µL of diluted streptavidin-PE was added to each well and the microparticle beads were incubated at RT for 30 min at 500 rpm on the orbital shaker. After a final 3 × 100 µL per well wash buffer on the vacuum manifold, the beads were resuspended with 100 µL wash buffer per well by incubation on the orbital shaker at 500 rpm for 2 min and immediately analysed on a Bio-Plex 200 System (Bio-Rad Laboratories Ltd, UK). Serum samples were diluted 1:2 with the calibrator diluent provided within the Luminex kit (LXSAMS-09: CCL2/MCP-1/JE; CCL3/MIP-1 alpha; CCL4/MIP-1 beta; CCL5/RANTES; CXCL1/GRO alpha/KC; CXCL10/IP-10; CXCL2/Gro beta/MIP-2/CINC-3; IL-6; TNF-alpha). Experimental samples were quantified against the standard for each cyto/chemokine. Briefly, for SAA3, experimental samples were quantified by ELISA against the SAA3 standard on a pre-coated plate containing anti-mouse SAA3 capture antibody and a paired anti-mouse SAA3 detection antibody in a standard

sandwich ELISA. Fold change of cyto/chemokine levels were calculated by dividing the mean values of proteins in immunisation groups by the mean values of proteins in the saline -alone group. The significance of the changes induced by different vaccines over those produced by saline alone were analysed using two-way ANOVA followed by Dunnett's multiple comparisons test. Correlations between the fold changes of transcripts and proteins were calculated using the Pearson correlation coefficient.

## Acknowledgements

This work was supported by a research program funded by a grant to RJS at Imperial College for the BioVacSafe project (Grant agreement number: 115308–2) that received support from a joint undertaking by the European Union's Seventh Framework Programme (FP7/2007-2013), the Innovative Medicines Initiative and in-kind contributions from EFPIA companies. SHEK acknowledges support from IMI JU Project 'BioVacSafe' (Grant No. 115308), a joint undertaking by the European Union's Seventh Framework Programme (FP72007-2013) and the Innovative Medicine Initiative. We thank Dr Hans-Joachim Mollenkopf and the microarray facility team at MPIIB for generating the transcriptomic expression data. We gratefully acknowledge Dormeur Investment Service Ltd for providing funds to purchase equipment used in these studies.

## Additional information

### Competing interests

Robert A van den Berg, Giuseppe Del Giudice: is an employee of the GSK group of companies. Reports ownership of shares and/or restricted shares in GSK. The other authors declare that no competing interests exist.

### Funding

| Funder | Grant reference number | Author |
|---|---|---|
| European Union Seventh Framework Programme | 115308-2 | Paul F McKay<br>Deniz Cizmeci<br>Yoann Aldon<br>Jeroen Maertzdorf<br>January Weiner<br>Stefan HE Kaufmann<br>David JM Lewis<br>Robert A van den Berg<br>Giuseppe Del Giudice<br>Robin J Shattock |
| Innovative Medicines Initiative | 11530 | Stefan HE Kaufmann |

The funders had no role in study design, data collection and interpretation, or the decision to submit the work for publication.

### Author contributions

Paul F McKay, Conceptualization, Data curation, Formal analysis, Supervision, Investigation, Visualization, Methodology, Writing—original draft, Project administration, Writing—review and editing; Deniz Cizmeci, Data curation, Software, Formal analysis, Visualization, Methodology, Writing—review and editing; Yoann Aldon, Investigation, Writing—review and editing; Jeroen Maertzdorf, Validation, Investigation; January Weiner, Robert A van den Berg, Formal analysis, Methodology, Writing—review and editing; Stefan HE Kaufmann, Resources, Writing—review and editing; David JM Lewis, Funding acquisition, Project administration, Writing—review and editing; Giuseppe Del Giudice, Writing—review and editing; Robin J Shattock, Conceptualization, Funding acquisition, Methodology, Writing—original draft, Project administration

## Author ORCIDs
Paul F McKay ⓘ https://orcid.org/0000-0001-5195-6254
Deniz Cizmeci ⓘ https://orcid.org/0000-0003-3231-7726
Yoann Aldon ⓘ https://orcid.org/0000-0001-9831-9391
January Weiner ⓘ https://orcid.org/0000-0003-1438-7819
Stefan HE Kaufmann ⓘ http://orcid.org/0000-0001-9866-8268

## Ethics

Animal experimentation: The animal studies were approved by the Ethical Review Board of Imperial College London where the experiments were carried out and work was performed in strict compliance with project and personal animal experimentation licences granted by the UK government in accordance with the Animals in Scientific Procedures Act (1986)- PPL 70-7457 Protocol #1. Animals received minimal handling and their physical condition was monitored at least twice daily. All procedures were performed under isoflurane anaesthesia when appropriate, and all efforts were made to minimise suffering. There was a detailed protocol in place, as per requirement of the humane endpoints described in the animal licence, for early euthanasia in the event of onset of illness or significant deterioration in condition. At the end of the experiment all animals were culled using a schedule 1 method and death confirmed before necropsy. Food and water were supplied ad libitum.

## Decision letter and Author response
Decision letter https://doi.org/10.7554/eLife.46149.019
Author response https://doi.org/10.7554/eLife.46149.020

# Additional files

## Supplementary files
• Transparent reporting form
DOI: https://doi.org/10.7554/eLife.46149.015

## Data availability

Complete microarray data was deposited in NCBI's Gene Expression Omnibus and is accessible through GEO accession number GSE120661.

The following dataset was generated:

| Author(s) | Year | Dataset title | Dataset URL | Database and Identifier |
|---|---|---|---|---|
| McKay PF, Cizmeci D | 2019 | Identification of Biomarkers of Vaccine Reactogenicity | https://www.ncbi.nlm.nih.gov/geo/query/acc.cgi?acc=GSE120661 | NCBI Gene Expression Omnibus, GSE120661 |

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
