## [Decision Letter]

Thank you for submitting your article "Identification of potential biomarkers of vaccin reactogenicity" for consideration by *eLife*. Your article has been reviewed by three peer reviewers, and the evaluation has been overseen by a Reviewing Editor and Arup Chakraborty as the Senior Editor. The following individual involved in review of your submission has agreed to reveal their identity: Ali Harandi (Reviewer #1).

The reviewers have discussed the reviews with one another and the Reviewing Editor has drafted this decision to help you prepare a revised submission.

Summary:

This well written paper by McKay and colleagues represents original research that focuses on evaluating the signatures of reactogenicity (adverse reaction to vaccination) to four different licensed vaccines, two adjuvants, and two TLR agonists in a F1 mouse model. The authors evaluated transcriptional profiles in the blood, draining lymph nodes, and the muscle at the injection site. The authors present the high level overview of the transcriptional responses on gene level (Figure 1), and on module level (Figure 2), and report different kinetics of responses to vaccines, adjuvant, and TLR agonists in the sampled tissues. They also show the clusters of temporally co-expressed genes by WGCNA and report overlaps between WGCNA clusters and BTM modules (Figure 3). In Figure 4, they describe the correlations in gene expression amongst different tissues, focusing on a small subset of soluble factors (10 genes), with high level of upregulation in all three tissues in response to injections. They also evaluated quantities of the 10 proteins in serum, where 9 out of these 10 factors are most strongly upregulated by LPS and Poly I:C at early (4-8 hr) time points following injection. In Figures 6 and 7 they report correlations between gene expression levels of these 10 factors and protein concentrations in serum. The experiments are well performed and the well presented data could be used as an important resource to predict or better understand the correlates/surrogates of vaccine reactogenicity/inflammation and safety.

Essential revisions:

Although the current version of the paper has several strengths, a few critical issues need to be addressed in a resubmitted version of the manuscript. In view of the observations presented in this manuscript, it has been suggested that the paper be submitted as a Tools and Resources manuscript.

1) The key objection expressed by the reviewers relates to the goal as stated in the manuscript, namely the developing a signature of vaccine reactogenicity. The reviewers opined that this is incorrect as reactogenicity per se was not measured. TLR agonists are indeed highly inflammatory and hence what the authors demonstrated are inflammatory responses. Also, the 10 gene signatures that the authors report are a signature of LPS and Poly I:C response. Therefore, it has been suggested that the title be revised to reflect that this study was done in mice and that the signature genes are not indicators of reactogenicity but rather inflammation: "Identification of potential biomarkers of vaccine inflammation in mice". In addition, the paper should be submitted as a Tools and Resources manuscript.

2) Please revise the text of the manuscript to reflect this change as well.

3) Please elaborate on several of the pathway/biomarkers in terms of their mechanistic contribution to vaccine efficacy, immunogenicity, etc. In the process, please elaborate on your observations vis-a-vis previously published literature on this subject, e.g., IFN-I response, chemokines, such as CXCL10.

---

## [Author Response]

Essential revisions:Although the current version of the paper has several strengths, a few critical issues need to be addressed in a resubmitted version of the manuscript. In view of the observations presented in this manuscript, it has been suggested that the paper be submitted as a Tools and Resources manuscript.1) The key objection expressed by the reviewers relates to the goal as stated in the manuscript, namely the developing a signature of vaccine reactogenicity. The reviewers opined that this is incorrect as reactogenicity per se was not measured. TLR agonists are indeed highly inflammatory and hence what the authors demonstrated are inflammatory responses. Also, the 10 gene signatures that the authors report are a signature of LPS and Poly I:C response. Therefore, it has been suggested that the title be revised to reflect that this study was done in mice and that the signature genes are not indicators of reactogenicity but rather inflammation: "Identification of potential biomarkers of vaccine inflammation in mice". In addition, the paper should be submitted as a Tools and Resources manuscript.

We thank the reviewers and editors for their careful assessment of the manuscript and agree that reactogenicity per se has not been measured and that we are using an inflammatory signal as a surrogate of reactogenicity. We tussled with how to describe our findings and are more than happy to use inflammation rather than reactogenicity to more properly describe the transcriptomic data. We are also happy to resubmit this paper as a Tools and Resources manuscript. We have changed the title as suggested.

2) Please revise the text of the manuscript to reflect this change as well.

We have revised the text of the paper as suggested.

3) Please elaborate on several of the pathway/biomarkers in terms of their mechanistic contribution to vaccine efficacy, immunogenicity, etc. In the process, please elaborate on your observations vis-a-vis previously published literature on this subject, e.g., IFN-I response, chemokines, such as CXCL10.

We have added in a discussion of the relevance of our identified chemokines in the context of vaccination as suggested elaborating on the role of cell recruitment, the interferon response and inflammation to the potential levels of immune responses.